# Locally orderless networks

Jon Sporring[*1,2], Peidi Xu[1], Jiahao Lu[1], François Lauze[1,2], and Sune Darkner[1]

[1]Department of Computer Science, University of Copenhagen, Denmark
[2]Center for Quantification of Imaging Data from MAX IV (QIM), Denmark
 {sporring, peidi, lu, francois, darkner}@di.ku.dk

## Abstract

We present Locally Orderless Networks (LON) and the theoretical foundation that links them to Convolutional Neural Networks (CNN), Scale-space histograms, and measure theory. The key elements are a regular sampling of the bias and the derivative of the activation function. We compare LON, CNN, and Scale-space histograms on prototypical single-layer networks. We show how LON and CNN can emulate each other, and how LON expands the set of functions computable to non-linear functions such as squaring. We demonstrate simple networks that illustrate the improved performance of LON over CNN on simple tasks for estimating the gradient magnitude squared, for regressing shape area and perimeter lengths, and for explainability of individual pixels' influence on the result.

## 1    Introduction

We introduce the locally orderless network (LON) which locally transforms the input signal into a set of local histograms. We propose a small modification of a standard convolutional neural network such that the internal representations becomes local histograms, as they have been studied in the classical scale-space literature [1]. This novel representation includes learnable operators which expands the representation space to a wide group of non-linear functions.

In [1], the authors presents the Locally Orderless Image (LOI) which identifies the 3 essential scale parameters of any gray-valued image: the resolution of the image, the size of the region of analysis, and intensity resolution. The authors show how these concepts are unified into a single scale-space paradigm, and at its heart resides local histograms which are the smooth and well-posed. With the strong foundation of classic scale space and measure theory [2, 3], in this article we illustrate how basic theoretical perspectives can be used to analyze these types of networks including classic concepts of invariance and density estimates. We focus on understanding the fundamental capacities of the LON and comparing it to Convolutional Neural Networks

(CNN) [4]. Thus we describe the LON from a theoretical perspective and compare it with a similar CNN on simple binary examples to illustrate some of their properties.

To the best of our knowledge, there is little prior work that is closely related to the LON. Histograms have been used in networks previously, where [5] introduced the histogram layer for more compact representation and realized that this is induced by the activation function. In [6] a global version was introduced, and [7] showed how such a representation can be used to estimate quantiles and subsequently distance. Density estimation deep learning network components exist, but most of that relates to density for loss functions such as cross-entropy. Pooling operation which can be interpreted as a histogram operation, and the learning of pooling operations was suggested by [8] where the pooling operation was learned during training. As an alternative one could consider Bayesian Neural Networks [9] that work with parameter uncertainty or distributions. In contrast, LON works with data distribution. There are however some works before the breakthrough of neural networks. In [10, 11], local histograms were used for texture classification with success. Furthermore, many of the classical image descriptors such as histogram of Gaussians (HoG), Daisy [12], and SIFT [13] are all based on histograms of features in some way.

## 2    Locally orderless histograms as convolutional networks

Consider an image and two kernels $I, K, W : \mathbb{R}^d \to \Gamma$, with support in $\Omega \subseteq \mathbb{R}^d$, and $\Gamma \subseteq \mathbb{R}$, a function $f : \mathbb{R} \to \mathbb{R}$, and scalars $b \in \mathbb{R}$, and the function $h : \Omega \times \mathbb{R} \to \mathbb{R}$,

$$h(\mathbf{x}, b) = \int_{\mathbb{R}^d} f\left(b - \int_{\mathbb{R}^d} I(\beta - \alpha)K(\alpha)d\alpha\right)W(\mathbf{x} - \beta)\,d\beta,\tag{1}$$

written as $h(\mathbf{x}, b) = (W * f(b - (I * K)))(\mathbf{x})$. When $W$ is a smoothing kernel, and $\int_{-\infty}^{s} f(t)dt$ is a sigmoid function, then $h(\mathbf{x}, b)$ is a local histogram value of $I * K$ in the neighborhood of position $\mathbf{x}$ and intensity $b$. In [1], $W$ and $K$ are Gaussian, $f$ is an unnormalized Gaussian with $f(0) = 1$, and all 3 functions had independent width parameters.

---

*Corresponding Author.

Proceedings of the 6th Northern Lights Deep Learning Conference (NLDL), PMLR 265, 2025.

We consider a discrete space $\Omega = \{\mathbf{x}_k\}_k \subset \mathbb{Z}^d$ and introduce a locally orderless network layer for discrete inputs $I_k = I(\mathbf{x}_k)$ as a linear combination of $M$ local histograms $h_j$ with individual kernels $K_j$ and $W_j$ and $2NM$ kernel-dependent intensities $b_{ij}$ and bell-shaped functions parametrized by $\sigma_{ij}$, as LON $: \mathbb{R}^{|\Theta|} \times \Gamma^{|\Omega|} \to \mathbb{R}^O$,

$$h_{ijk} = \left(W_j * f_{\sigma_{ij}}(b_{ij} - (I * K_j))\right)(\mathbf{x}_k), \quad (2)$$
$$\text{LON}\left(\{I_k\}\right) = \mathbf{A}\,\text{vec}(\{h_{ijk}\}) \quad (3)$$

where $\Theta = \{K_j, b_{ij}, \sigma_{ij}, W_j, \mathbf{A} : i = 1 \ldots N, j = 1 \ldots M\}$ is the set of parameters, $\mathbf{A} \in \mathbb{R}^{MN|\Omega| \times O}$ is a matrix and $O$ is the number of output connections under suitable boundary conditions for the convolution operator, and $|\cdot|$ is the cardinality operator.

In the following, we will give an interpretation of LON by comparing it with a similar convolutional network.

## 2.1 Circumference versus areas

LON and convolutional networks (CNN) compare as the circumference and area of objects. A CNN similar to (3) is found by replacing the bell-shaped functions $f_{ij}$ with a single sigmoid function $g_{ij}(v) = \int_{-\infty}^{v} f_{ij}(w)\,dw$,

$$\text{CNN}(\{I_k\}) \quad (4)$$
$$= \mathbf{A}\,\text{vec}(\{(W_j * g_{ij}(b_{ij} - (I * K_j)))(\mathbf{x}_k)\}).$$

Consider a family of activity functions in LON which converges to Kronecker's delta function $f_{\sigma_{ij}} \to \delta$, as $\sigma_{ij} \to 0$, then correspondingly, the activity functions in CNN will converge to the Heaviside function, $g_{ij} \to H$. As a consequence,

$$f_{ij}(b_{ij} - (I * K_j)(\mathbf{x})) \to \begin{cases} 1, & b_{ij} - (I * K_j)(\mathbf{x}) = 0 \\ 0, & \text{otherwise,} \end{cases} \quad (5)$$

$$g_{ij}(b_{ij} - (I * K_j)(\mathbf{x})) \to \begin{cases} 1, & b_{ij} - (I * K_j)(\mathbf{x}) \geq 0 \\ 0, & \text{otherwise.} \end{cases} \quad (6)$$

Thus LON focuses on the isophotes of $b_{ij} - (I * K_j)(\mathbf{x})$, while CNN performs a threshold of the same term. When $f$ has a finite width, then LON defines well-posed, soft isophotes. Given a connected, compact region $S \subset \Omega$ and an image $I = \chi_S + \varepsilon$, where $\chi_S$ is the indicator function and $\varepsilon$ is i.i.d. noise, the circumference and area of S is,

$$\text{Circum}(S) \sim \sum_k \left(\delta * f(0.5 - (I * K))\right)(\mathbf{x}_k), \quad (7)$$

$$\text{Area}(S) \sim \sum_k \left(\delta * g(0.5 - (I * K))\right)(\mathbf{x}_k), \quad (8)$$

where $K$ is a smoothing kernel, and $\mathbf{A}$ are implied to be a 1-vector with $|\Omega|$ elements.

## 2.2 LON and nonlinear measures

LON expresses some local operators more naturally than CNN. For a transformation $\xi : \Gamma \to \Gamma$, $J(\mathbf{x}) = \xi(I(\mathbf{x}))$, and for a probability mass function $h_I$, by the Law of the unconscious statistician (LUS) we have,

$$\mathbb{E}\left(J\right) = \sum_\Gamma \xi(i) h_I(i). \quad (9)$$

A local version is obtained by convolution with a smoothing kernel $W$,

$$\left(W * J\right)(\mathbf{x}) \simeq \sum_{i \in \Gamma} \xi(i) h(\mathbf{x}, b_i), \quad (10)$$

since the above is linear in $h$ then it can be written on the form (3). Further, any linear combinations of transformations $\xi_m : \Gamma \to \Gamma$,

$$\sum_m \left(W * J_m\right)(\mathbf{x}) \simeq \sum_m \sum_{i \in \Gamma} \xi_m(i) h(\mathbf{x}, b_i). \quad (11)$$

is also linear in $h$ and thus can also be written on the form (3).

As an example, consider derivative kernels, $K_k(\mathbf{x})$ where $\mathbf{x} = (x_1, x_2, \ldots, x_d)$, and such that $I_k = I * K_k$ is a smoothed estimate of the directional derivative in the $x_k$-direction. With $\xi(v) = v^2$, the LON can approximate the gradient magnitude squared as,

$$\text{grad}^2(\mathbf{x}) \simeq \sum_{k=1}^d \left(W * I_k^2\right)(\mathbf{x}) \simeq \sum_{k=1}^d \sum_{i \in \Gamma} i^2 h_k(\mathbf{x}, i), \quad (12)$$

Note that for linear functions $\xi$ and Gaussian kernels $W$ and $K$ or its derivatives with standard deviation $\gamma$ and $\sigma$, then convolution semi-group property of Gaussian kernels implies that the two kernels can be replaced with a single Gaussian kernel of standard deviation $\sqrt{\gamma^2 + \sigma^2}$ with an appropriate sum of their derivative orders. This does not hold when $\xi$ is non-linear, but our experience is that the resulting scale of $\text{grad}^2$ is close to $\sqrt{\gamma^2 + \sigma^2}$.

## 3 Experiments

In this section, we will compare LON (3) with CNN (5). To focus on the inner parts of the networks, we set $W_j = \delta$, where $\delta$ is the Dirac delta function, and we investigate their ability to estimate the gradient magnitude squared, to estimate and classify circumferences and areas of objects, and we perform a sensitivity analysis.

## 3.1 The gradient magnitude squared

The gradient magnitude squared is a rotational invariant indicator of the apparent edge of object

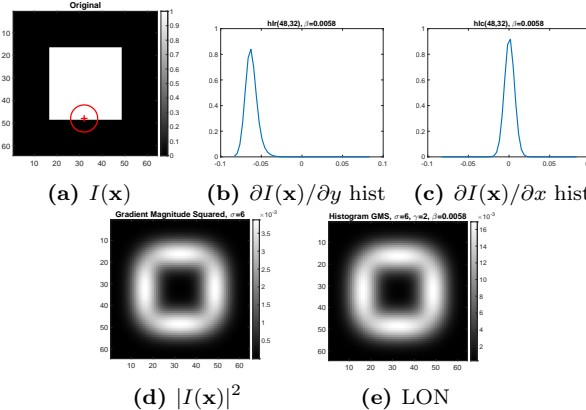

**(a)** $I(\mathbf{x})$    **(b)** $\partial I(\mathbf{x})/\partial y$ hist    **(c)** $\partial I(\mathbf{x})/\partial x$ hist

**(d)** $|I(\mathbf{x})|^2$    **(e)** LON

**Figure 1.** Comparing LON with the gradient magnitude. The local histogram of the vertical and horizontal derivative are shown for the point and extend indicated by the red graphics in (a), and the locally orderless network (12) with $A \sim \{i^2\}$.

parts. Using spatial coordinates $\mathbf{x} = [x, y]$, a direct implementation is given by $|I(\mathbf{x})|^2 = (\partial I(\mathbf{x})/\partial x)^2 + (\partial I(\mathbf{x})/\partial y)^2$. Examples of $|I(\mathbf{x})|^2$ and (12) on a simple image are shown in Figure 1. For both experiments, $K$ and $W$ were Gaussian kernels with standard deviation 6 and 2 respectively with infinite extend. We see that LON produces results that are visually very similar to the direct implementation of the gradient magnitude.

To compare the ability of CNN and LON to learn the gradient magnitude squared, we have calculated the gradient magnitude squared using $|I(\mathbf{x})|^2$ for objects from the MNIST database [14]. Our hypothesis is that for a two-kernel system, both CNN and LON will learn a set of orthogonal derivative kernels, but in contrast to LON, CNN will not be able to learn the square nature of the gradient magnitude. To highlight the difference, the intensities of each hand-written character are multiplied by a random scalar sampled from the continuous uniform distribution from 0.5 to 2.0, examples of which are shown in Figure 2(a) and Figure 2(b). The networks all consist of a $3 \times 3$ convolution with their respective activation functions, followed by a $1 \times 1$ convolution. This results in networks with $21 - 35$ parameters in total. All networks are trained for 2000 epochs with batch size 2048, using Adam with a learning rate 0.005 and pixel-wise mean squared error. All models converged fast with similar learning curves. Examples of results for both CNN and LON are shown in Figure 2 for a varying number of kernels, $f$, and the number of regular samples on the bias axis $i$. For both the CNN experiments, we see that edges at certain angles are not modeled correctly, while LON with 2 kernels successfully captures the edges in all directions, and LON with 8 kernels also captures the intensity variation accurately as reflected in the loss. The CNN with sigmoid activation appears to

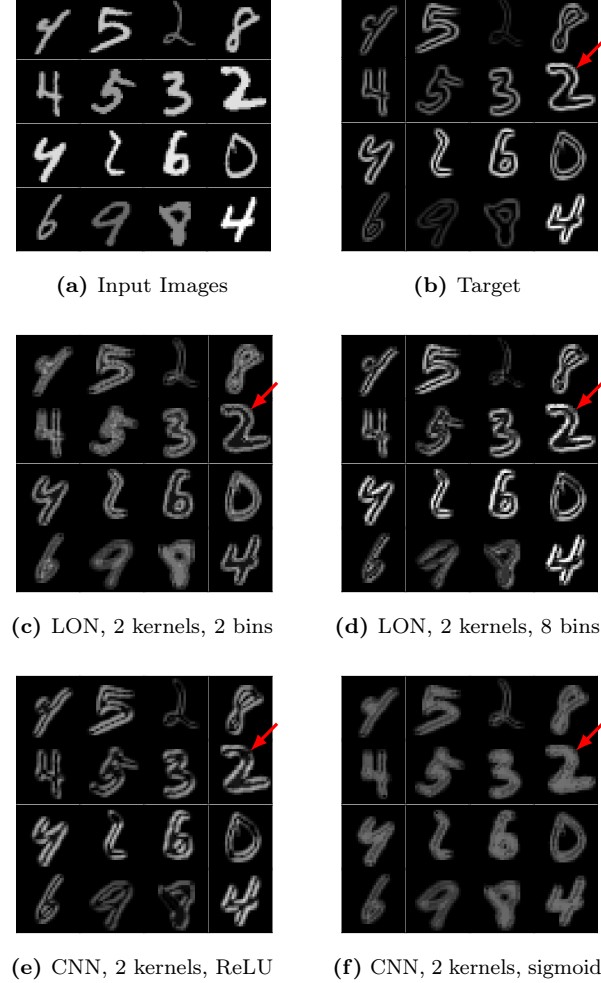

**(a)** Input Images    **(b)** Target

**(c)** LON, 2 kernels, 2 bins    **(d)** LON, 2 kernels, 8 bins

**(e)** CNN, 2 kernels, ReLU    **(f)** CNN, 2 kernels, sigmoid

**Figure 2.** CNN achieves similar behavior as LON, but while LON succeeds in all directions, CNN does not.

have significant difficulties with intensities and fails to predict the edges, particularly at certain orientations, (example is indicated by the red arrow in figure 2)

## 3.2 Circumference versus area

We hypothesize that the structure of CNN will excel, when working with areas in $K * I$, due to its ReLU activation function, while LON will excel for isophotes in $K * I$, since it essentially operates on histogram bins. To test this, we constructed a stochastic source of objects with varying area and perimeter, by generating random $512 \times 512$ images from identically and independently distributed (iid) normal noise. A Gaussian filtering was then applied with $\sigma = 10$. The foreground areas are then selected as having intensity values $> 75\%$ quantile. With the thresholded image, we then run connected component decomposition to separate each random shape while ignoring the incomplete shapes near boundaries as well as too-small or too-large shapes. We finally place such a shape into an image of fixed size $128 \times 128$. The

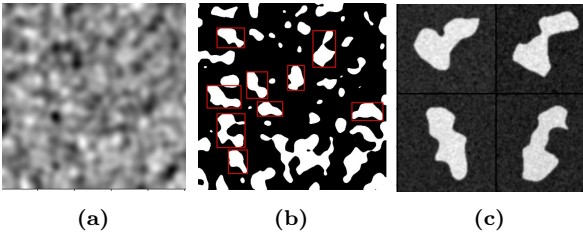

**(a)**        **(b)**        **(c)**

**Figure 3.** Process for random shape generation: An image of iid normal noise smoothed with a Gaussian kernel (a), its threshold and similar components (b), and exemplar objects with added iid noise (c).

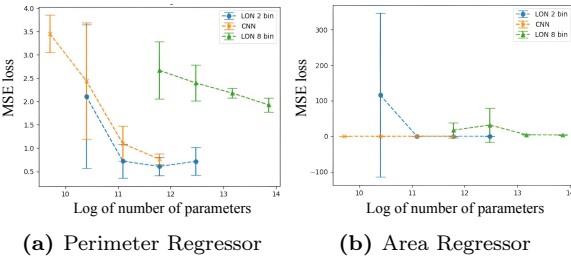

**(a)** Perimeter Regressor      **(b)** Area Regressor

**Figure 4.** The mean square error by the logarithm of the number of parameters for CNN with 2 kernels and the ReLU, LON with 2 kernels and 2 bins and 8 bins on the regression task on the perimeter and area of random shapes without iid noise (Figure 3). Lower is better.

process is illustrated in Figure 3.

For the regression task, we tested networks with varying numbers of kernels, and the results are shown in Figure 4 on noiseless images of random objects. Note that the number of parameters varies for CNN and LON by the number of kernels and bins. With an image with $|\Omega|$ pixels, $M$ kernels of $|K|$ pixels, CNN has $M|\Omega| + M|K|$ parameters. On the other hand, LON further has $N$ bins and thus $NM|\Omega| + M|K|$ parameters. In direct comparison, the number of parameters is dominated by the $|\Omega|$, and hence, LON is $N$ times larger than a CNN with the same number of kernels. However, if the CNN is given $NM$ kernels to compare with a LON with $N$ bins and $M$ kernels, then the LON has $(N-1)M|K|$ fewer parameters. In our experiments, we compared the network's performance on a logarithmic scale, where the subtle difference in the number of parameters is not highlighted. The models were trained using Adam optimizer, and we tuned the learning rate of both $1 \cdot 10^{-3}$ and $5 \cdot 10^{-4}$ for experiments, and only the best results are reported.

For the regression task, we see wrt. the perimeter, the LON outperforms CNN in the 2-bin case in terms of the number of parameters used, however, for the 8-bin case it seems that the LON overfits. Wrt. area, LON with 2 bins has trouble converging, while CNN outperforms both LON.

For the classification task, we divided the objects into small, medium, and large wrt. either the area or perimeter length, and asked the network to correctly

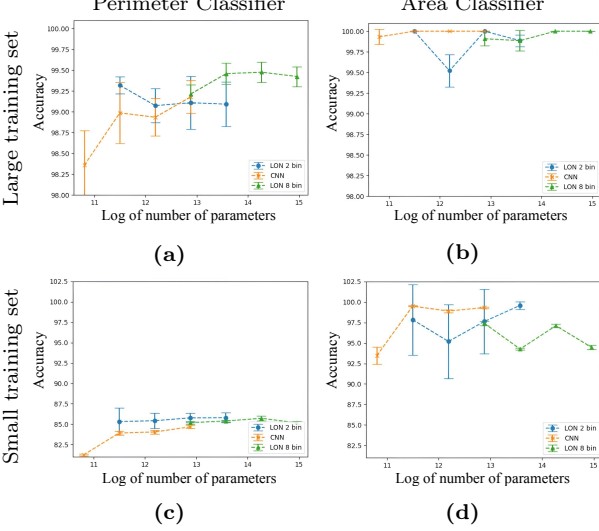

**(a)**          **(b)**

**(c)**          **(d)**

**Figure 5.** The accuracy by the logarithm of the number of parameters for CNN with 2 kernels and the ReLU, LON with 2 kernels and 2 bins and 8 bins on the regression task on the perimeter and area of random shapes without noise but trained on many ($\approx$4000) or few ($\approx$1500) examples (Figure 3). Higher is better.

classify the three classes either by perimeter or area, while keeping the other constant. We also explored this problem in terms of a large or small training set. The results as a function of the number of parameters are shown in Figure 5. Again it appears that LON with 2 bins is better at classifying objects in terms of their perimeter, while CNN is better at classifying objects wrt. area.

## 3.3 Explainability by sensitivity maps

Explainability is an increasingly important property of machine learning algorithms, and as LON is linked to the boundary between apparent object parts, we hypothesize that sensitivity maps for LON will be more meaningful than those of a CNN. We define sensitivity maps as

$$\left| \frac{\partial E}{\partial I} \right| = \left| \frac{\partial E(Y, L(I))}{\partial I} \right| \quad (13)$$

where $E$ is the error or loss function, $Y$ is the true class, $L$ is the network, and $I$ is the input image. The sensitivity maps express the gradient of each pixel wrt. to the similarity, thus what change in the similarity a change in pixel value will cause. These are often considered the features of interest.

In this experiment, we consider the classification task on random, noise-free shapes, shown in Figure 3(c). The resulting sensitivity image is shown for various combinations of networks, channels, and the essential number of bins in Figure 6. It shows

| Type | Kernels | Perimeter classifier (constant area) | | | Area classifier (constant perimeter) | | |
|---|---|---|---|---|---|---|---|
| CNN | 2 | pred=0, true=0 | pred=1, true=1 | pred=2, true=2 | pred=0, true=0 | pred=1, true=1 | pred=2, true=2 |
| | 8 | pred=0, true=0 | pred=1, true=1 | pred=2, true=2 | pred=0, true=0 | pred=1, true=1 | pred=2, true=2 |
| LON 2 bins | 2 | pred=0, true=0 | pred=1, true=1 | pred=2, true=2 | pred=0, true=0 | pred=1, true=1 | pred=2, true=2 |
| | 8 | pred=0, true=0 | pred=1, true=1 | pred=2, true=2 | pred=0, true=0 | pred=1, true=1 | pred=2, true=2 |
| LON 8 bins | 2 | pred=0, true=0 | pred=1, true=1 | pred=2, true=2 | pred=0, true=0 | pred=1, true=1 | pred=2, true=2 |
| | 8 | pred=0, true=0 | pred=1, true=1 | pred=2, true=2 | pred=0, true=0 | pred=1, true=1 | pred=2, true=2 |

**Figure 6.** Comparing sensitivity maps for perimeter and area classification on noise-free images (Figure 3). Shapes are grouped into 3 classes (pred=0, 1, 2, denoting small, medium, large, respectively). All networks use 2 kernels and light pixels have a large influence on the classification accuracy.

that although all the models can make correct predictions with very high accuracy, as demonstrated in Figure 5(a) and Figure 5(b), the sensitivity of the two models focusing on totally different regions. LON looks into the boundaries of the shape to get a perimeter estimator, especially with 8 bins where CNN on the other hand is inferring the perimeter from both the foreground and the background.

For the case of area classification, the sensitivity map of CNN is still quite noisy and somewhat inconsistent in contrast to the LON which begins to make predictions based on the boundaries but also takes the inside into account with more kernels or more bins. It seems that the sensitivity maps for LON are far more consistent across variations in parameters than for CNN and far better aligned with what humans would perceive as important for the two tasks.

# 4  Conclusion

In this article, we have explored the relationship between local histograms and CNNs. At the core, we have shown that by changing the activation function from a sigmoid to its derivative, the internal representation becomes similar to histograms. With further simple requirements on the kernels, we arrive at the classical scale-space histogram of locally orderless images. We call networks with such layers for Locally Orderless Networks (LONs). In this article, we focus on some theoretical insights to be gained. We have shown how LONs extend the set of simply representable functions with non-linear functions as squaring, and how a LON can be designed to express a CNN and vice-versa. LONs excel in capturing both rotational invariance and amplitude in tasks like gradient magnitude estimation. Finally,

we have presented an initial set of experiments to study the empirical qualities of LON as compared to CNN, which suggest that LONs outperform CNNs in tasks involving boundaries, indicating improved explainability. Future work will explore combining LONs with existing networks, particularly for segmentation tasks.

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
