# OpenReview forum: "Locally orderless networks"
_NLDL.org/2025/Conference — NLDL 2025 Oral_

### Official Review · Reviewer_cJRh · 2024-09-26
**Locally Orderless Networks - Interesting approach with narrow / limited results**

**Confidence:** 4

**Summary:**

The article introduces Locally Orderless Networks; operators with roots in Locally Orderless Image frameworks. The work draws on kernel methods via general Parzen-Rosenblatt windows and local histograms, and connect these to the general convolutional operator from CNNs. The link is well grounded in theory, and the authors reference earlier work with local feature descriptors (SIFT, HoG).

Compared to CNNs, LON emphasizes soft isophotes (boundaries of regions of constant value), whereas CNNs perform thresholding. This can potentially make LON effective in certain tasks where CNNs might theoretically be less well suited, such as shape regression. The authors also investigate LONs for gradient based saliency maps.

**Strengths:**

(S1) The general idea behind LONs is quite intuitive and well grounded in theory.

(S2) LONs could potentially be used to extract robust locally permutation invariant features, complementing existing methods.

(S3) LONs provide novelty, and the theoretical justification for their application is well motivated in the manuscript. There is an obvious, but appealing connection to existing convolutional operators.

(S4) The presentation is generally sound, concise, and precise.

**Weaknesses:**

(W1) The experiments are limited in scope, and its general applicability to more complex or diverse tasks remains mostly unexplored. It is not clear from the experiments that LONs offer significant benefits in general modelling settings. This is by far the most pressing weakness of the work, as it provides little incentive for direct applications.

(W2) It seems that LON requires more parameters compared to CNNs, especially as the number of bins increases and particularly in the area regressor experiment. While this is somewhat discussed, the current results seem to indicate issues for LONs with higher number of bins. As a result, the reader is left with more questions regarding when exactly this operator could serve as viable modelling tool.

(W3) While the saliency maps provided by LONs are more focused, the definition of saliency with pure gradients is not considered robust. In this reviewers opinion, it somewhat demonstrates that LONs seems well suited to shape regression. However, pure gradients as a general saliency method has been demonstrated to be unreliable in more general settings [(Adebayo et al. 2018)](https://arxiv.org/abs/1810.03292).

**Final Rebuttal Confidence:**

5

**Final Rebuttal Justification:**

We appreciate that the authors are looking to establish the feasibility of a more theoretical approach. We believe the idea could have merit, as discussed in our review, however, the experiments designed around the current approach does not clearly show or delineate significant benefits of the proposed approach.

In summary, we believe the rebuttal fails to sufficiently establish the value of the proposed LON network layer in a modern modeling framework. It is still not sufficiently clear to the reader how LONs provide benefits compared to a standard convolutional layer, and the authors comments are not sufficiently addressing this limitation. For this reason, we will maintain our recommendation to not accept the work as part of the conference.

We wish the authors luck in their future work towards locally orderless network layers.

**Justification:**

**Strengths**: LONs provide an intuitive method that could likely be purposeful for more effectively extracting certain locally permutation invariant features. In particular, LONs seem promising for extracting statistical features that could require multiple spatial / temporal CNN layers to adequately model. It would be interesting to see LONs compared more directly as an adaptive variant to classic feature extraction methods such as SIFT and HoG; a somewhat missed opportunity in this reviewers opinion. This could also potentially remediate certain issues with the current draft, as the current experiments are a little too narrow in scope to convincingly argue for their practical use in modern vision modelling tasks, see below for details.

**Weaknesses**: Overall; while LONs presents an interesting theoretical and practical tool, they still have challenges regarding generalisation, parameter efficiency, and scalability. Moreover, the combination of histogram-based features with kernel methods, especially RBFs, has been widely studied and applied, and while LON can be seen as a continuation or extension of this approach within the neural network paradigm, the authors should perhaps more extensively acknowledge the body of existing research that intersects with its core concepts. This, in conjunction with slightly more extensive experiments, would strengthen the manuscript substantially. Leaning into a more theoretical / mathematical result for justifying the class of problems where LONs can extract meaningful features could potentially provide a more convincing argument for the method.

**Summary**: While the method is novel, the article is missing that little extra push to make this reviewer confident of the contribution of LONs for the general community. Currently, this reviewer is left with more questions after reading the manuscript than before reading it. Consider adding more experiments to advocate for exactly where practitioners may apply LONs in a modelling context, perhaps by formulating LONs in context of adaptive local feature descriptors, or perhaps as a non-parametric tool with Gaussian Mixtures or Dirichlet processes.

---

> ### Author Rebuttal · Authors · 2024-10-24
>
> Thank you for your insightful and constructive feedback. We appreciate your acknowledgment of the theoretical grounding of our work and the potential of LONs. The editors asked us to address specific questions, so we respond to your identified weaknesses for further discussions:
>
> (W1) The experiments are limited in scope, and …
>
> A1: We present a theoretical investigation into a simple change of standard convolutional networks, which allows for the calculation of nonlinear functions. We emphasise theory since to improve the balance between application and theory in the field of neural networks, and presently space and time did not permit an in-depth investigation into more than superficial applications.
>
> (W2) It seems that LON requires more parameters compared to CNNs, …
>
> A2: Overfitting for LON with 8 bins only happens in perimeter regressor and when using small training set for classification. However, with sufficient training data, the model outperforms others (despite having a large number of parameters) for the perimeter classification task, as shown in Figure 5 (a). LON with 8 bins also displays a clearer boundary contour in Figure 6 (the saliency map) for the perimeter classification, highlighting its ability to capture and represent the relevant features more effectively when enough data is available.
>
> (W3) While the saliency maps provided by LONs are more focused, …
>
> A3: We have removed the term saliency map and used sensitivity which is well known from statistics. This in principle tells us the impact of changing the input space on the output and is the correct description of the analysis.

---

### Official Review · Reviewer_rzBp · 2024-10-09
**Locally Orderless Networks as a generalization of  Convolutional Neural Networks**

**Confidence:** 2

**Summary:**

This paper introduces a novel neural network layer called Locally Orderless Networks (LON) and explores its theoretical connections with Convolutional Neural Networks (CNN) and scale-space histograms. The core of LON lies in the regular sampling of the bias and the derivative of the activation function to create local histograms. By comparing LON, CNN, and scale-space histograms on prototypical single-layer networks, the paper demonstrates LON's advantages in specific tasks such as gradient magnitude squared estimation, shape area, and perimeter length regression, as well as explainability of individual pixel influence on the results.

**Strengths:**

The paper's strengths include its solid theoretical foundation linking Locally Orderless Networks (LON) with established concepts like CNNs and scale-space histograms, the introduction of an innovative neural network layer that enhances model capabilities, superior performance in tasks requiring boundary recognition and rotational invariance demonstrated through rigorous experimental comparison with CNNs, improved model explainability which is essential for trust and transparency in AI, and thorough experimental validation that substantiates LON's effectiveness across various tasks.

**Weaknesses:**

The weaknesses of the paper encompass the increased complexity and computational demands of LON due to its higher number of parameters compared to CNNs, concerns about its generalization ability beyond the specific tasks and datasets explored, the potential for overfitting highlighted by certain experiments, and the need for broader experimental validation to ascertain LON's versatility and practicality in diverse applications.

**Final Rebuttal Confidence:**

2

**Final Rebuttal Justification:**

It is difficult for me to assess whether the strengths and weaknesses of the model presented in this paper are critical.

**Justification:**

The introduction of a new model is certainly commendable. However, as I am not an expert in modeling, it is difficult for me to assess whether the strengths and weaknesses of the model presented in this paper are critical.

---

> ### Author Rebuttal · Authors · 2024-10-24
>
> Thank you for your feedback and for recognizing the theoretical contributions of our work, as well as the improvements in model performance and explainability offered by LON. The editors asked us to address specific questions, so we have decided to respond to your identified weaknesses as question:
>
> W1: The weaknesses of the paper encompass …
>
> A1: We too are concerned with overfitting. We have used early stopping to limit the risk of overfitting, and the indicated underperformance discussed wrt. Figure 4a could indicate that we could have performed better with a larger training set. We will look into this in the future.

---

### Official Review · Reviewer_xnJP · 2024-10-09
**Locally Orderless Networks - Strengths in theoretical foundations and weaknesses in experimental scope and generalizability**

**Confidence:** 4

**Summary:**

The paper "Locally Orderless Networks" (LONs) introduces a novel neural network architecture that integrates scale-space theory and measure theory, enhancing the computational capabilities of neural networks by employing locally orderless histograms. It establishes LONs as a generalization of the Locally Orderless Image framework, allowing for the computation of non-linear functions like squaring. The research demonstrates that LONs outperform Convolutional Neural Networks (CNNs) in tasks related to edge detection, gradient magnitude estimation, and shape analysis, showcasing superior performance in object classification based on perimeter and improved explainability through saliency maps. Overall, LONs offer a theoretically sound and effective alternative to CNNs, particularly for boundary-focused tasks, with potential for further integration in future work.

**Strengths:**

The paper presents a well-founded proposal that integrates scale-space theory and measure theory, demonstrating how LONs generalize the locally orderless image paradigm to compute non-linear functions such as squaring. Empirical evaluations show that LONs outperform CNNs on specific tasks that require nuanced spatial and intensity handling, such as gradient magnitude estimation and shape analysis. The explainability analysis via saliency maps also reveals improvements over CNNs, indicating enhanced explainability by better highlighting task-relevant areas. Furthermore, it underscores the potential applicability of combining LONs with existing architectures in future research. Overall, this work represents a valuable contribution to the field, effectively illustrating the potential of LONs across several domains.

**Weaknesses:**

1. The experiments primarily focus on gradient magnitude estimation and shape analysis, lacking a broader range of tasks and datasets. Expanding the experimental scope would strengthen claims about the versatility and generalizability of LONs.

2. The paper compares LONs with CNNs and does not include a broader analysis against other state-of-the-art architectures.

3. While the experiments indicate that LONs may exhibit overfitting, the paper does not adequately address methods to mitigate this issue. Incorporating techniques like regularization or cross-validation would enhance the reliability of the findings.

4. The paper lacks a thorough discussion on the scalability and computational efficiency of LONs, particularly regarding the added parameters from locally orderless histograms.

5. Although the theoretical foundations are solid, the paper does not explore the limitations and assumptions of LONs in depth. Understanding the conditions under which LONs might fail or produce suboptimal results would enhance credibility.

6. Certain terminology and concepts, such as scale-space and measure theory, should be better explained.

7. The font size of Figures 1, 4, and 5 is very small and cannot be read.

**Final Rebuttal Confidence:**

4

**Final Rebuttal Justification:**

The authors have satisfactorily addressed all of my comments and have made appropriate changes to strengthen the critical aspects of the paper. Therefore, my recommendation is to accept the paper for publication.

**Justification:**

While the paper has notable strengths, it also has several weaknesses that should be addressed before publication. Although the theoretical foundations are sound, the limited experimental scope raises concerns about the generalizability of the findings. Additionally, the lack of rigorous comparative analysis diminishes the paper's significance. It would also be beneficial to discuss potential overfitting, as well as issues related to scalability and computational complexity. Finally, while the paper is generally clear, some terminology may pose challenges for readers unfamiliar with the underlying theories, and certain figures need correction. Addressing these weaknesses could enhance the paper's overall impact and relevance.

---

> ### Author Rebuttal · Authors · 2024-10-24
>
> Thank you for your thoughtful and detailed feedback. We appreciate your recognition of the solid theoretical foundations of our work. We have revised the manuscript to improve the clarity of the terminology, and enhanced the readability of the figures as suggested. The editors asked us to address specific questions, so we have decided to respond to your identified weaknesses as question:
>
> W1: The experiments primarily focus …
>
> A1 We understand the concern of the reviewer, however, there is a trade-off between a theoretical analysis of our LONs and the place left for experiments. The page limitation makes it very complicated, as with a new type of layer, some theoretical analysis seems necessary to explain this type of construction (rephrase!)
>
> W2: The paper compares LONs with CNNs and does not include a broader analysis against other state-of-the-art architectures.
>
> A2: We appreciate the suggestion and will include it in an extension of the paper. Unfortunately the space available for this paper is too short for including it here.
>
> W3: While the experiments indicate that …
>
> A3: Since our purpose is to study the effects of the histogram layer and standard convolutional layer, we avoid other tricks like regularization for a pure comparison of layer operations. Cross-validation is not used for any parameter tuning, but the results of figure 4 and 5 illustrate the mean and std of model performances from a 5-fold cross-validation.
>
> W4: The paper lacks a thorough discussion on …
>
> A4: It is correct that we do not evaluate the computational complexity of the method, but we do, in Section 3.2, give a general equation for the number of parameters given the input size etc. We also discuss the tradeoff between the LON’s number of parameters and the functions computable by it.
>
> W5: Although the theoretical foundations are solid, …
>
> A5: This is a very good question, which most likely is valid for most of the deep learning methods used today. We will look into this in the future, but an in-depth discussion requires much more space than the 6 pages available for us here.
>
> W6: Certain terminology and concepts, such as scale-space and measure theory, should be better explained.
>
> A6: This is a very important point and central to understanding in general how convolutional networks function. With the limited space available, we have added intuition about these two general concepts.
>
> W7: The font size of Figures 1, 4, and 5 is very small and cannot be read.
>
> A7: We have increased the font size.

---

### Official Review · Reviewer_qFuz · 2024-10-09
**Locally Orderless Networks**

**Confidence:** 4

**Summary:**

The paper proposes an architecture called "Locally Orderless Networks" (LONs), which is analogous to CNNs with biases but uses Gaussian functions as activations. The authors note that sigmoid functions, which are common activations in CNNs, can be viewed as integrals of Gaussian functions. Therefore, while a CNN layer performs soft thresholding of the convolved input, a LON layer implements a "soft indicator" of the convolved input. According to the authors, this property makes LONs a better choice for tasks, which involve estimating lengths of curves of the same intensity in images (such as perimeter estimation), while CNNs are more appropriate for area estimation tasks. The paper provides experiments to support this intuition, where LONs marginally outperform CNNs for perimeter estimation. Moreover, the authors suggest that LONs result in more intuitive saliency maps in comparison with CNNs, and thus provide better explainability.

**Strengths:**

**S1**: The main idea is simple, clear, and intuitive.

**S2**: Authors propose a concrete area, where LONs can be applied (tasks involving perimeter estimation in images).

**S3**: The figures are overall nicely designed, and the experiments illustrate the paper's message well.

**Weaknesses:**

**W1**: The novelty of the paper is marginal, given that the LON architecture amounts to simply using a different activation function in a CNN.

**W2**:  The writing could be significantly improved, especially in the introduction and conclusion.

**Questions/suggestions**:
* **Q1**: In the proposed architecture, the outer layer has the number of parameters proportional to the image size $|\Omega|$. I.e., every spatial location of the convolved image is multiplied by an independent set of weights. However, since the goal of the main experiments is just to estimate the perimeter or area, it should not be important where exactly the object appears in the picture. Therefore, shouldn't it be sufficient to use the same parameters for all the spatial locations (i.e., use a variant of a convolution in the outer layer)? Why do the authors choose to train seemingly much more parameters than needed for these tasks?
* **Q2**: Which exact LON is displayed in Figure 1? What are the kernels parameters?
* Figures 4 and 5 would benefit from bigger legend and font sizes.
* **Q3**: Figure 6 only shows correctly classified samples. How would missclassified samples look?
* Classes «0,1,2» should be defined in Figure 6 (I assume they correspond to area/parameter classes from the smallest to the largest?).
* **Q4**: How do the authors interpret the behaviour of LON with 8 bins and 2 kernels in Figure 6? How does it relate to the mentioned "overfitting" in Figure 4? Is it related to the much larger parameters count than needed (see **Q1**)?

**Writing problems/typos**:
* The dimensions of $A$ in line 089 should be transposed.
* There are many unclear/undefined wording choices in the introduction. E.g.: "function width" (I assume it is $\sigma$, but the word choice in confusing in NNs context), "local histogram", "scale space", "activity/activation" (are these used interchangeably?), "sigmoid derivatives» (is this the same as Gaussian functions?).
* In my view, the locally orderless images framework and its relationship to the proposed architecture is not explored and explained enough in the text. However, I understand that it is partially due to the limited space.
* There are some language mistakes. E.g., "network components exists" (line 049), "LON expresses more naturally to some local operators than CNN" (line 123), etc.
* The conclusion is especially poorly written and includes sentences that are incorrect and misleading. E.g., "... layer, called Locally Orderless Networks (LONs), allows CNNs and LONs to model each other and compute non-linear functions, like squaring".

**Final Rebuttal Confidence:**

4

**Final Rebuttal Justification:**

Since I already recommended acceptance in the initial review, there is no change in my assessment post rebuttal.

**Justification:**

Since I could not find any major issues with soundness and correctness of the paper, and given strengths S1 and S2, I believe that the paper's contribution is worth sharing with the community. While the novelty and originality of the paper are not very high (W1), I believe that the work is carefully done and is worth publishing after some minor revision of the writing (W2).

---

> ### Author Rebuttal · Authors · 2024-10-24
>
> Thank you for your input, we have improved the writing quality as guided by yours and the other reviewers suggestions. Also, we thank you for noting that although a minor change of the general network structure, the theoretical consequence is noteworthy.
>
> Q1: In the proposed architecture, …
>
> A1: The number of parameters mostly comes from the last fully connected layer when the output feature map from CNN/LON has been flattened, which is why it is dependent on the image size. Since our purpose is to compare the operation of a single-layer CNN and single-layer LON with few kernels, flattening the output layer is the most feasible way, allowing for a fair comparison in terms of the features they learn and the patterns they can recognize. On the other hand, although location-invariant operations like Global average pooling can remove the dependency on the image size, it would produce insufficient features for single-layer network structures.
>
> Q2: Which exact LON is displayed in Figure 1? What are the kernel parameters?
>
> . Figures 4 and 5 would benefit from bigger legend and font sizes.
>
> A2.1: The LON for the gradient magnitude (figure 1) task is defined in eq 12, where the h function is defined in eq. 2. Unlike the kernels to be learned (optimised) for circumference and area task, here the two convolutional kernels W and K are just pre-defined Gaussian kernels with given standard deviations, as mentioned in the paragraph below eq 12, which are used to compute the gradient magnitude. We realised that we did not give the specific numbers for the standard deviations. We have now added a sentence in Section 3.1 saying that "K and W were Gaussian kernels with standard deviation 6 and 2 respectively".
>
> A2.2: Fonts in Figures 4 and 5 have been enlarged.
>
> Q3: Figure 6 only shows correctly classified samples. How would missclassified samples look?
>
> . Classes «0,1,2» should be defined in Figure 6 (I assume they correspond to area/parameter classes from the smallest to the largest?).
>
> A3.1: Our purpose is on the theoretical analysis of networks, thus it is more interesting to examine what CNNs and LONs are looking at when they make correct predictions (in most cases). Showing misclassified examples would introduce noise and highlight edge cases that do not represent typical model behaviour. Given that misclassifications are outliers and not reflective of the network’s general operation, they are less relevant to our study.
>
> A3.2: Yes! We have added a note in the figure captions saying "pred=0, 1, 2, denoting small, medium, large, respectively".
>
> Q4: How do the authors interpret …
>
> A4: The regression task is not completely consistent with the classification task. As can also be seen in Figure 5, for classification tasks, overfitting for LON with 8 bins only appears when using a small training set.  However, with sufficient training data, LON with 8 bins outperforms others (despite having a large number of parameters) for the perimeter classification task. This is why the LON with 8 bins displays a clearer boundary contour in Figure 6 (the saliency map) for the perimeter classification, highlighting its ability to capture and represent the relevant features more effectively when enough training data is available.

---

### Meta-Review · Area_Chair_h5iB · 2024-11-01

**Recommendation:** Accept (Poster)
**Confidence:** 4

**Metareview:**

The reviewers agree that the paper is well-motivated and has a good theoretical foundation. Reviewers, however, also differ in their opinions, with two positive reviewers, one neutral, and one that recommends a rejection. The strong aspects of the paper are its theoretical foundation, that it is based on a clear, intuitive, and simple idea, and that it is well described. The highlighted weaknesses are limitations of the experimental validation, limited scope of the method, limited novelty, and insufficient discussion of computational complexity, scalability, and generalizability.

The authors have given a thorough rebuttal that addresses all comments from the reviewers and rewritten the paper accordingly. So, with the primarily positive reviews, I recommend accepting the paper as a poster.

**Suggested Changes To The Recommendation:**

3: I agree that the recommendation could be moved up

---

### Decision · Program_Chairs · 2024-11-06

**Decision:**

Accept (Oral)

**Comment:**

Given the AC positive recommendation, we recommend an oral and a poster presentation given the AC and reviewers recommendations.